# The Sexual and Parenting Rights of People with Physical and Psychical Disabilities: Attitudes of Italians and Socio-Demographic Factors Involved in Recognition and Denial

**DOI:** 10.3390/ijerph19021017

**Published:** 2022-01-17

**Authors:** Simona Gabriella Di Santo, Margherita Colombo, Marco Silvaggi, Giorgia Rosamaria Gammino, Valentina Fava, Chiara Malandrino, Chiara Nanini, Cristina Rossetto, Sara Simone, Stefano Eleuteri

**Affiliations:** 1IRCCS Fondazione Santa Lucia, Via Ardeatina 306, 00179 Roma, Italy; 2Italian Association of Applied Sexology and Psychology (AISPA), Via Marostica 35, 20146 Milano, Italy; 3Superior School of Clinical Sexology of Turin, Via Unione Sovietica 335, 10135 Torino, Italy; 4Institute of Clinical Sexology (ISC), Via Savoia 78, 00198 Roma, Italy; m.silvaggi@gmail.com; 5Italian Center of Sexology (CIS), Via Scipione Dal Ferro, 4, 40138 Bologna, Italy; g.r.gammino@gmail.com; 6Research Group for Sexology, Via S. Sofia 78, 95123 Catania, Italy; valentinafavamp4@yahoo.it (V.F.); chiara.malandrino@yahoo.it (C.M.); 7Interdisciplinary Centre for Research and Training in Sexology (CIRS), Via Angelo Ceppi di Bairolo 1/8, 16126 Genova, Italy; chiarananini@virgilio.it; 8Study Center for Affective and Sexual Disorders Treatment (DAS), Via G.T. Invrea 20/2, 16129 Genova, Italy; cristinarossetto@icloud.com; 9Institute of Research and Training (IRF), Via Luigi Alamanni 23, 50123 Firenze, Italy; sarasimone77@hotmail.com; 10World Association of Sexual Health, c/o Kristen Mark World Association for Sexual Health Secretariat 1300 S. 2nd Street, Suite 180, Minneapolis, MN 55454, USA; stefano.eleuteri@uniroma1.it; 11Faculty of Medicine and Psychology, Sapienza University of Rome, Via di Grottarossa 1035, 00185 Roma, Italy

**Keywords:** sexual rights, physical disabilities, psychical disabilities, sexual minorities, societal attitudes

## Abstract

The sexual and parenting rights (SPRs) of people with disabilities (PwDs) are under-recognized. Sociodemographic factors may influence attitudes towards them. The aims of this study were: (1) to analyze the levels of agreement of a sample of Italian people with some SPRs of PwDs; (2) to inquire if the SPRs of people with psychical disabilities (PwPSYDs) were less recognized than those with physical disabilities (PwPHDs); (3) to verify if sociodemographic characteristics associated with under-recognition. An online anonymous survey was distributed using non-random sampling methods to conduct an inquiry into the level of agreement with statements regarding the SPRs of PwPHDs and PwPSYDs to have satisfying sexuality, to marry, and to adopt children. Answers from 973 Italian participants, aged 18–84 years (71.1% females) were analyzed. At least 70% of respondents declared in favor of the SPRs of PwPHDs. The SPRs of PwPSYDs were always subjected to higher under-recognition. Religiosity, male sex, higher age, and lower education were the factors most often associated with being against the SPRs of PwDs. Improved identification of the less tolerant respondents and the less recognized categories may allow for specific strategies for promoting the recognition of the SPRs for PwDs.

## 1. Introduction

More than 15% of the world’s population lives with one or more disabilities, including physical and sensory impairments, developmental and intellectual disabilities, and psychosocial disabilities [1]. Disabilities are defined by the UN Convention on the Rights of Persons with Disabilities (UNCRPD) as the presence of *long-term physical, mental, intellectual, or sensory impairments, which in interaction with various barriers may hinder their full and effective participation in society on an equal basis with others* [2]. Disabilities, therefore, are not caused by the impairments themselves, but rather by the presence of barriers in the physical environment, access to information and education, laws and norms, services, and societal beliefs and behaviors [3].

One main barrier that people with disabilities (PwDs)—and in particular those with psychical and intellectual impairments—are still forced to face, concerns the full realization of their sexual and parenting rights (SPRs). SPRs are today considered fundamental human rights, and their violation constitutes a violation of the rights to equality, non-discrimination, dignity, and health [4]. The SPRs of PwDs concerning marriage, family, parenthood, relationships, fertility, access to information, and sexual and reproductive health services, and freedom from exploitation and abuse are supported at the institutional level [1,2] Despite this, the sexual rights of people with disabilities struggle to be recognized in societies, often due prejudices and stereotypes. Stereotypes associated with PwDs presume that they are childlike, not sexually attractive, with underdeveloped or abnormal sexual desires, and maladjusted to any kind of sexual expression [5]. “Myths” about disability include prevalent and related false beliefs, such as the ‘myth of physical perfection’ [6] which considers PwDs as not sexually attractive; the ‘myth of asexuality’ [7,8], which elicit perceptions of vulnerability, childlike innocence, and dependency, and justifies the presumption of an absence of romantic and erotic experiences in PwDs, who are seen as holy innocents disinterested in sex; the ‘fucking ideology’ [7], which considers ‘sex’ synonymous with heterosexual penetrative intercourse with male-dominant sex positions, and it makes it difficult to imagine PwDs able to have sex, because of their impairments. It is hypothesized that these myths and stereotypes influence the attitudes of people in general and, specifically, of parents or caregivers who care for PwDs. As a result, PwDs face a myriad of demand and supply-side barriers to accessing sexual and reproductive healthcare [9,10].

Evidence also shows that attitudes towards PwDs also differ by type of disability, with those with more visible disabilities [3] or people with psychical disabilities (PwPSYDs) often facing greater discrimination than persons with physical disabilities (PwPHDs) [11]. PwPSYDs are deemed unable to provide valid consent [12], to marry or have children [13], or, far from being angelic, to be hypersexual, unable to control themselves, devoid of any inhibition, irresponsible, and sometimes perverse [14]. Sexual stigma disfavors opportunities for intimate relationships, procreation, sexual education, and sexual health. Sexuality is often discouraged and inaccessible to many adults with disabilities. PwDs often lack safe, private places to engage in partnered or individual sexual activities, and reproductive and parenting rights of PwDs, and in particular for PwPSYDs, are often viewed negatively by family members, service providers, and the general community [15,16]. Moreover, PwDs are particularly at risk of coercion or of undergoing sterilization or contraceptive procedures, such as intrauterine device (IUD) insertion without their free and fully informed choice and consent. This is especially the case for women with major or multiple impairments or with PSYDs [17]

Some authors have tried to inquire which personal characteristics of the population affect attitudes regarding sexuality in PwDs, with inconsistent results, plausibly due to methodological and sample differences. Some studies suggest an effect of age, with older people expressing fewer accepting positions [18,19,20], while others have failed to find an association between acceptance and age [21,22]. Similarly, while some evidence exists that females displayed more positive attitudes related to sex and disability [22,23], others revealed that males held more liberal opinions [24], and one showed no relationship between gender and attitude towards sexuality [20]. As regards education, some studies revealed an association between higher levels of instruction and more positive attitudes [25], while others showed no relationship [21,22,26]. Attitudes towards sexuality in PwDs may also vary due to beliefs associated with the cultural origin [3,27,28] and religion with more religious people tending to hold more conservative opinions [25]. However, most research on attitudes towards the SPRs of PwDs has engaged primarily their family members, supporting staff, or university students, while a few updated studies enrolled samples from the general population [3,27,29].

In Italy, serious issues of prejudice and discrimination against minorities exist, to the point that a bill was recently proposed in Parliament seeking to punish acts of discrimination and incitement to violence against gay, lesbian, transgender people, and PwDs [30]. The so-called “Zan Law”, after being approved in the Chamber of Deputies, was recently rejected in the Senate [31]. This refusal immediately provoked several street demonstrations in the main Italian squares, in which thousands of people took part [32]. Indeed, even if, apparently, Italians would be mostly in favor of granting more rights to minorities, our previous study also indicated that SPRs are less recognized in minorities, such as LGB people [33]. As far as we know, no recent research in Italy inquired into the societal level of agreement with the SPRs of PwPHDs and PwPSYD, or the socio-demographic predictors of unfavorable attitudes towards reducing inequalities.

The aim of this study was therefore to analyze the level of agreement of the Italian general population with the right of PwPHDs and PwPSYDs to have satisfactory sexuality, to marry and to adopt a child, and to evaluate whether the SPRs of PwPSYDs were subject to less recognition, compared to PwPHDs. A further objective was to better understand which socio-demographic characteristics of Italian people were significantly associated with a lower agreement with the SPRs for PwPHDS and PwPSYDs.

## 2. Materials and Methods

### 2.1. Inclusion/Exclusion Criteria for Study Participants

The study was designed to be proposed to the entire Italian adult population. Therefore, criteria for participation were having reached the age of majority, under Italian laws, and being born and resident in Italy.

### 2.2. Study Design and General Description

The data reported in this paper is extracted from a larger, original study which, through an online questionnaire, cross-sectionally investigated the attitudes of Italian people regarding the right of particular minorities or demographic categories (i.e., heterosexual, homosexual, bisexual and transsexual men, women or couples; PwPHDs; PwPSYDs, minors; seniors; sex-workers) to have satisfying sexuality, to marry, to adopt children, to be hired for any job, and to be free to choose how they lived with their sexuality [34].

The study was approved by the Scientific Committee of the Italian Federation of Scientific Sexology (FISS) and carried out in cooperation between the Youth Section of FISS (FISS Youth) and the Youth Initiative Committee of the World Association of Sexual Health (WAS YIC).

### 2.3. Specific Contents of the Questionnaire

The online questionnaire was developed with Google Forms (Google LLC, Mountain View, CA, USA) to collect information related to socio-demographic data of respondents and their level of agreement with statements related to the abovementioned SPRs.

Two main kinds of information were considered:

(a) Socio-demographic information: age in years (categorized in three groups of comparable size); biological sex (‘female’, ‘male’); education (categorical question, which included all obtainable levels of Italian education and which was recoded into ‘undergraduate’, ‘graduate’, and ‘post-graduate’); region of origin (open question, which was recoded in coming from ‘North’, ‘Center’ or ‘South-Islands’); occupational status (‘student’, ‘employed’, or ‘unemployed/retired’); occupation (open question, which was recoded in ‘healthcare worker’ including doctors, psychologists, nurses and other health professionals and ‘other profession’), sexual orientation (completely heterosexual, mostly heterosexual, bisexual, mostly homosexual and completely homosexual, which was recoded in ‘completely heterosexual’ or ‘not-completely heterosexual’); relational status (open question, which was recoded in ‘single’, ‘in a committed relationship’, ‘cohabiting with a partner’, ‘married’); religiosity (‘non-believer’, ‘believer’, ‘practicing’); kind of religion (multiple choice with free field to enter non-prefilled data).

(b) The level of agreement with the right for PwPHDs, and PwPSYDs to have satisfying sexuality, to marry, and to adopt children, coded on a 6-point Likert-type scale, (1 = complete disagreement; 2 = moderate disagreement; 3 = mild disagreement; 4 = mild agreement; 5 = moderate agreement; 6 = complete agreement). To prompt to a stance, it was not possible to express intermediate opinions between agreement and disagreement or to avoid answering one or more questions.

### 2.4. Data Collection

The questionnaire was distributed via email and through the main social media by all the authors and by colleagues affiliated with their associations, through an exponential non-discriminative snowball sampling approach. A direct link to the questionnaire was also published on the website of the Italian Federation of Scientific Sexology (www.fissonline.it, accessed on 28 September 2021). The first page of the form presented the proponent institutions, explained the rationale, objectives, and contents of the survey, and advised that the survey was reserved exclusively for people aged 18 or over. ‘I agree’ or ‘I do not agree’ options on the first page of the questionnaire were used in place of a signature, to provide informed consent.

People who did not accept to participate were redirected to the last page of the form and thanked for their potential interest. In this case, no data, except that a refusal had occurred, together with its date of occurrence, was recorded. People who accepted to participate accessed the survey. In the second case, all their answers were automatically collected by the software.

### 2.5. Statistical Analyses

All data transformations and analyses were performed with IBM SPSS version 20 (SPSS Inc., Chicago, IL, USA). Descriptive analyses included calculation of the mean ± standard deviations for continuous variables and absolute frequencies and percentages for categorical ones. The Shapiro–Wilk test was performed to evaluate the normality of distributions.

Since answers to most questions about SPRs were highly skewed towards higher scores, the Wilcoxon matched-pairs test was used to compare, the levels of agreement with the right of PwPHDs and PwPSYDs to each right.

Multiple ordinal regression models failed to respect the assumption of proportional odds, due to a large number of empty cells. Therefore, answers to questions related to the SPRs were dichotomized (a score of 1, 2 or 3 was classified as “disagreement” and a score of 4, 5, 6 as “agreement”). Then, univariate and multiple binary logistic regression analyses were conducted to identify the socio-demographic variables associated with “agreement” with each SPRs for PwPHDs and PwPSYDs, taking “disagreement” as a reference. Values were expressed as odds ratio (OR) and 95% confidence intervals (95% CI). To exclude multicollinearity, the variance inflation factor (VIF) was calculated for each predictor. Dummy variables were obtained from non-dichotomous ones and calculated by selecting for reference the most prevalent category. A VIF greater than 2.5 (which corresponds to an R² of 0.60) was considered indicative of multicollinearity.

The level of significance was established at 95% (*p*  <  0.05) for all statistical analyses.

## 3. Results

### 3.1. Descriptives

One thousand and seven out of 1015 people who had accessed the questionnaire provided informed consent to the research. Of these, 34 (3.4%) were excluded from further analyses because they were not born or did not live in Italy.

The analyzed sample was therefore composed of 973 participants, aged between 18 and 84 years (mean age: 35.5 ± 11.7; median age: 33.0). Demographics are shown in detail in Table 1.

Most participants were female (71.1%) and people under 40 years of age (70.5%); middle-aged people and seniors (people over 50) represented 12.4% of the sample. Six-hundred and forty-one (65.9%) were graduates and post-graduates; 21.9% of the sample consisted of healthcare workers (34.3% of employed). Four-hundred and eighty-two out of 509 participants declared themselves non-atheists were Catholics (94.6%); therefore, due to the low presence of non-Catholic responders, ‘kind of religion’ was excluded from potential predictors of ‘agreement’.

### 3.2. The Right of PwPHDs and PwPSYDs to Have Satisfactory Sexuality

#### 3.2.1. Percentages of Agreement and Comparison between PwPHDs and PwPSYDs

Overall, more than 85% of the sample declared in agreement with the right of PwDs to have satisfactory sexuality (Figure 1). Only 1.1%, 1.4%, and 2.4% of participants answered being completely, moderately, or mildly against it, while 4.1%, 8.9%, and 82.0% declared mildly, moderately, or completely in favor of it. A slightly lower number of respondents expressed complete, moderate, or slight agreement with the right of PwPSYDs to have satisfactory sexuality (63.4%, 11.4%, and 10.8%, respectively), while 7.9%, 2.8%, and 3.7% were mildly, moderately, or completely opposed.

The Wilcoxon Signed-Rank test indicated that the median ranks for PwPHDs were statistically significantly higher than the median ranks for PwPSYDs (Z = 13.04, *p* < 0.001).

#### 3.2.2. Socio-Demographic Variables Associated with Agreement with PwPHDs’s Right to Have Satisfactory Sexuality

The results of the univariate logistic regression analyses suggested the existence of an association between demographics, such as sex, education, occupation, and relational status, and agreement with the SR of people with PHD to have satisfactory sexuality (Table 2). None of the VIF exceeded 2.5. The multiple logistic regression analysis included all predictors and indicated that only positive associations between ‘agreement’ and sex were maintained when taking into account other potential predictors. Women agreed in significantly higher percentages than men (96.1% vs. 92.5%) with the right of PwPHDs to have satisfying sexuality (O.R. = 2.11; 95% C.I.: 1.11–4.01; *p* = 0.022).

#### 3.2.3. Socio-Demographic Variables Associated with Agreement with PwPSYDs’s Right to Have Satisfactory Sexuality

The results of the univariate logistic regression analyses suggested the existence of an association between agreement with the right of people with PSYDs to have satisfactory sexuality and sex, religion, regional origin, occupational status, relational condition, and level of education (Table 2). None of the VIF exceeded 2.5. The multiple logistic regression analysis included all predictors and indicated that most of these associations maintained statistical significance. In particular, being female, atheist, a graduate, employed, or a student, and living with a partner were related to higher odds for agreement than being male (O.R. = 1.55; 95% C.I.: 1.03–2.35; *p* = 0.038), believer (O.R. = 1.82; 95% C.I.: 1.18–2.82; *p* = 0.007) or churchgoer (O.R. = 2.25; 95% C.I.: 1.3–3.89; *p* = 0.004), undergraduate (O.R. = 1.73; 95% C.I.: 1.1–2.72; *p* = 0.019), unemployed (employed: O.R. = 1.92; 95% C.I.: 1.18–3.11; *p* = 0.008; student: O.R. = 2.47; 95% C.I.: 1.27–4.81; *p* = 0.008), or single (cohabitant: O.R. = 1.92; 95% C.I.: 1.06–3.48; *p* = 0.032; married: O.R. = 1.87; 95% C.I.: 1.11–3.17; *p* = 0.019), respectively, while regional origin resulted in an association that was bordering on statistical significance (O.R. = 1.59; 95% C.I.: 1–2.54; *p* = 0.051).

### 3.3. The Right of PwPHDs and PwPSYDs to Get Married

#### 3.3.1. Percentages of Agreement and Comparison between PwPHDs and PwPSYDs

Overall, 98.3% of participants declared in agreement with the right of PwPHDs to marry, while 78.7% of them answered being favorable to marriage for PwPSYD (Figure 1). In particular, 88.8% of participants expressed strong agreement with the possibility for PwHDs to get married, while 6.7% and 2.8% declared moderately and mildly in favor, and only 1.7% expressed disagreement. A noticeable lower number of respondents expressed complete, moderate, or slight agreement with the right of PwPSYDs to marry (51.1%, 11.9%, and 15.7%, respectively), while 11.3%, 4.7%, and 5.2% were mildly, moderately, or completely opposed (Figure 1).

The Wilcoxon Signed-Rank test indicated that the median ranks for PwPHDs were statistically significantly higher than the median ranks for PwPSYDs (Z = 17.85, *p* < 0.001).

#### 3.3.2. Socio-Demographic Variables Associated with Agreement with PwPHDs’s Right to Get Married

The results of the logistic regression analyses are to be interpreted with caution, due to the extremely low number of people against marriage for PwPHDs in our sample (Table 3). Atheists/agnostics agreed in significantly higher percentages than churchgoers (99.6% vs. 95.9%) with the right of PwPHDs to marry, and a positive significant association seemed to exist between ‘agreement’ and religiosity in multiple regression analysis (O.R. = 7.22; 95% C.I.: 1.32–39.62; *p* = 0.023). 

#### 3.3.3. Socio-Demographic Variables Associated with Agreement with PwPSYDs’s Right to Get Married

The results of the univariate logistic regression analyses suggested the existence of an association between agreement with the right of people with PSYDs to marry and all considered predictors, with the exception of provenance (Table 3). None of the VIF exceeded 2.5. The multiple logistic regression analysis included all predictors and indicated that most of these associations maintained statistical significance in the multivariate model. In particular, being female, atheist/agnostic, post-graduate, student and non (completely) heterosexual were related to higher odds for agreement than being male (O.R. = 1.83; 95% C.I.: 1.28–2.63; *p* = 0.001), believer (O.R. = 1.47; 95% C.I.: 1.01–2.16; *p* = 0.046) or churchgoer (O.R. = 2.62; 95% C.I.: 1.65–4.17; *p* < 0.001), undergraduate (O.R. = 1.9; 95% C.I.: 1.21–2.98; *p* = 0.005), unemployed (O.R. = 2.31; 95% C.I.: 1.2–4.43; *p* = 0.012), or heterosexual (O.R. = 1.81; 95% C.I.: 1.13–2.88; *p* = 0.013).

### 3.4. The Right of PwPHDs and PwPSYDs to Adopt a Child

#### 3.4.1. Percentages of Agreement and Comparison between PwPHDs and PwPSYDs

The right of PwDs to adopt a child encountered the agreement of 71.7% of the sample, while 21.8% of respondents declared themselves in favor of the adoption of a child by the PwPSYDs (Figure 1). A total of 5.1%, 3.1%, and 10.1% of participants answered being completely, moderately, or mildly opposed to this right for PwPHDs, while 14.7%, 17.9%, and 49.1% declared mildly, moderately, or completely in favor. A noticeable lower number of respondents expressed complete, moderate, or slight agreement with the right of PwPSYDs adopt (6.0%, 3.5%, and 12.3%, respectively), while 21.1%, 19.7%, and 37.4% were mildly, moderately, or completely opposed.

The Wilcoxon Signed-Ranks test indicated that the median ranks for PwPHDs were statistically significantly higher than the median ranks for PwPSYDs (Z = 24.57, *p* < 0.001).

#### 3.4.2. Socio-Demographic Variables Associated with Agreement with PwPHDs’s Right to Adopt a Child

The results of the univariate logistic regression analyses suggested the existence of an association between education, religiosity, sexual orientation, and relational status, and agreement with the right of people with PHD to adopt a child (Table 4). In the multiple logistic regression analysis, indicated that positive associations were maintained between ‘agreement’ and education and sexual orientation when taking into account other potential predictors. Post-graduates and non (completely) heterosexuals agreed in significantly higher percentages than undergraduates (O.R. = 1.84; 95% C.I.: 1.14–2.98; *p* = 0.013) and heterosexual people (O.R. = 2.03; 95% C.I.: 1.26–3.27; *p* = 0.004) with the right of PwPHDs to adopt a child.

#### 3.4.3. Socio-Demographic Variables Associated with Agreement with PwPSYDs’s Right to Adopt a Child

The results of the univariate logistic regression analyses suggested the existence of an association between agreement with the right of people with PSYDs to adopt a child and age, sexual orientation religion, and level of education (Table 4). None of the VIF exceeded 2.5. The multiple logistic regression analysis included all predictors and indicated that religion and education maintained statistical significance in the multivariate model. Indeed, being post-graduate and atheist/agnostic were related to higher odds for agreement than being under-graduate (O.R. = 1.97; 95% C.I.: 1.26–3.08; *p* = 0.003), or churchgoer (O.R. = 2.28; 95% C.I.: 1.3–3.99; *p* = 0.004). 

## 4. Discussion

To our knowledge, this is the first published research to conduct an inquiry into the levels of agreement with the SPRs of PwDs in the Italian general population and the socio-demographic variables, which may associate with positive or negative attitudes.

The results of this study indicate that the very large majority of Italians declared in agreement with the right of PwPHDs to have satisfactory sexuality and to marry, and were most inclined to agree with their right to adopt a child, albeit to a lesser extent. Consistent with other research, the percentage of people who disagreed with these SPRs when related to PwPSYDs was significantly higher [11]. In particular, almost 80% of our sample declared against the right for adoption for PwPSYDs. Indeed, the sexuality of PwPSYDs was found to be judged as less appropriate than that of persons without any disability or with physical disabilities [11]. It was observed that stereotypes exist regarding PwPSYDs, who are deemed incapable of judgment and control, unpredictable, and possibly dangerous: these stereotypes may elicit false beliefs about unsafe sex, promiscuity, or abuse [11]. Even though we did not investigate beliefs or stereotypes associated with the recognition of the SPRs of PwDs, we can hypothesize that they may also explain our findings.

Consistent with other studies [18,19], opinions concerning the possibility for PwDs to adopt a child were more unfavorable than those related to sexuality and marriage, confirming that participants considered this aspect to be more complex and, perhaps, to have implications beyond the mere SPRs of the individuals, and to require abilities that many would judge to be too complex to be carried on by PwDs, and, in particular, by PwPSYDs. Different from another study by Cuskelly and Gilmore [18], who observed less favorable, yet still substantially positive, attitudes about parenting in PwPSYDs, Italian respondents declared for the most tendentially or completely against PwPSYDs being allowed to adopt a child. This difference could also be amplified by the fact that the present research requires a specific opinion related to adoption and not on the general possibility of being parents.

Consistent with other research, women appeared to be more supportive of PwDs’ right to have satisfactory sexuality and of PwPSYDs’ right to marry than men, while no differences between genders were observed about parenting [22,23].

Regarding socio-demographic variables, our data revealed that religion was a strong predictor of negative attitudes in most analyses, with churchgoers as the most uncompromising category. Indeed, religious culture, with a “religious ethical model” has contributed in the past to the structuring of “false myths” about disability, supporting the opinion that a disabled body or mind is an indication of an inner (moral) ugliness (sin); therefore, the myth of physical perfection that recalls purity and divine goodness, of very ancient memory, is very often found at the basis of prejudices on the causes of the impairment [6].

We also found a relationship between the level of education and the presence of agreement. In general, we observed that undergraduates tended to express more disagreement than graduates and post-graduates with SPRs of PwDs, and that post-graduates tended to agree in slightly higher percentages than graduates (with the exception of items relating to satisfactory sexuality). Other research revealed that students and professionals who had frequent contact with PwPSYDs generally had more negative attitudes compared to those who had less experience with them [35]. Even if in our sample medical doctors and psychologists represented up to 44.4% of post-graduates, our data appear to be inconsistent with the results of this research. However, it should be noted that our survey did not require participants to specify whether their work involved contact with (or care of) PwDs. For this reason, it is possible that our sample included both people who had frequent contact with PwDs and people who had rare or absent relationships with them.

In some comparisons, we observed that the proportion of people in agreement with the SPRs of the PwDs decreased with increasing age. However, in most univariate and multivariate analyses, age did not appear to be a significant predictor of agreement, contrary to what has been reported in some literature, which indicated age as an important predictor of negative attitudes towards the sexuality of PwD and, in particular, of those with intellectual disabilities. In any case, it is necessary to consider that our sample was mainly constituted by young people. Therefore, middle-aged and elderly people could be under-represented.

Sexual orientation was significant in predicting attitudes towards marriage and adoption, with not completely heterosexuals agreeing in a significantly higher percentage than heterosexuals (88.5% vs. 75.5%) to these SPRs. To our knowledge, there are no other studies that have investigated sexual orientation as a predictor for agreement with the SPRs of PwDs. We can hypothesize that, given that in the period in which the survey was administered, the question of the “Zan” law was much debated, non-heterosexual people were also much more sensitive to the issues of legitimizing unconventional couples or parenting for minorities.

Our study presents some strengths and many limitations. As abovementioned, the main limitations concern the poor representativeness of the enrolled sample, being composed mainly of women, young people, and graduates, and in high percentages by healthcare workers, therefore limiting the validity of this research. In any case, most of the literature on this topic derives from studies enrolling limited convenience samples of a few tens or hundreds of participants, mostly recruited among relatives, students, or care providers of PwDs. For this reason, even if not perfectly representative of the Italian population, our data fit into a line of research whose results tend to be hampered by selection bias.

Other limitations concern having included a limited number of variables among predictors. It is possible to hypothesize that people’s level of agreement may be associated with other factors, e.g., having a relative with physical/psychical disability, or pending an adoption. We can also hypothesize that people were more sensitized to these topics since they were often discussed in politics and media. However, it is not straightforward that greater exposure may have implied greater agreement.

A strength of our research is the fact that it is the first Italian study to have evaluated the opinions of people in the community regarding the civil and sexual rights of PwDs. Until now, to our knowledge, there were no Italian data relating to the topic.

Another strength is represented by the sample size. A sample of almost a thousand people is numerically very consistent and definitely higher than most of the works published so far on the topic. Although the results on the levels of agreement expressed by the entire sample must be interpreted with caution due to the aforementioned limitations, the sample size allowed to make predictions about the effect of different socio-demographic variables on the agreement with the SPRs of PwDs. Our data may therefore represent a useful background for more rigorous studies and interventions aimed at information and training campaigns on the issues of civil and sexual rights of minorities.

## 5. Conclusions

Despite its limitations, this study has important implications regarding the SPRs of PwDs in Italy. Firstly, even if quite positive opinions were expressed in most answers, important work is needed to overcome the cultural barriers that prevent PwDs from fully exercising their rights, in particular the right to adopt a child. Second, our data confirmed that PwPSYDs face more discrimination than PwPHIDs. Actions are needed to overcome the prejudice and stigma towards them. Lastly, evidence is shown that certain characteristics of the population affect the level of recognition of PwDs’ rights, in particular having strong religious beliefs is related to lower recognition. Furthermore, male sex, higher age, and lower education contributed in some analyses to outline the more uncompromising profiles. Other research demonstrated that specific training on sex and disability can lead to more positive attitudes related to their SPRs [22]. For this reason, it is necessary to work specifically on populations with the highest risk factors for uncompromising attitudes, focusing efforts where possible on improving knowledge and overcoming stereotypes.

People’s psychological well-being is severely hampered by a discriminatory environment. In Italy, ignorance and discrimination are also sustained by the lack of national policies of sexual education, and by the intention of the political majority to discourage any action that could reduce inequalities in SPRs or promote structured interventions of sex education at a national level. This is due to both cultural and economic reasons. The recent rejection of the ‘Zan Law’ [31] confirms that it is no longer possible to wait for the ideal solution represented by national policies, and it is important to design targeted interventions to optimize resources. To do this, we believe that the results of this study are useful because they indicate which people discriminate the most and which rights are denied the most. This can direct educational interventions on specific issues and aimed at proper recipients and conducted in proper environments, i.e., where people are most discriminated against.

## Figures and Tables

**Figure 1 ijerph-19-01017-f001:**
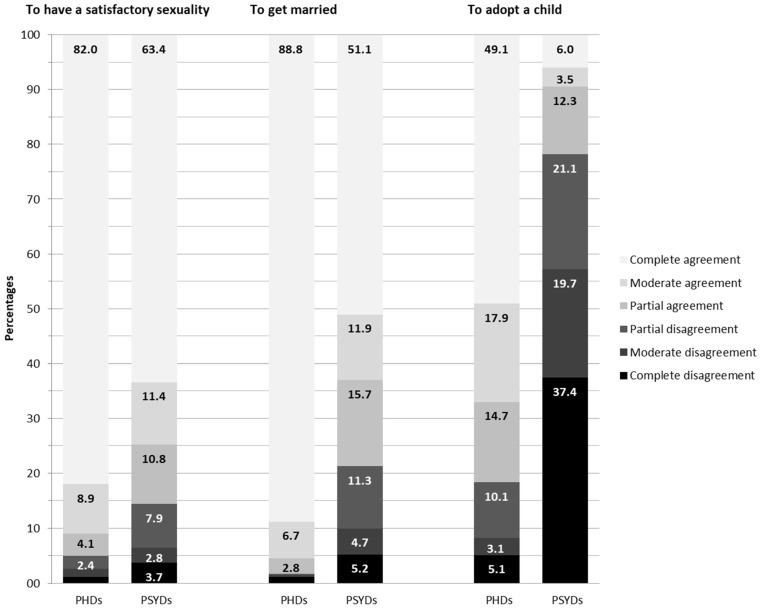
Distribution of responses inquiring agreement with the sexual rights of people with physical disabilities (PHDs) and psychical disabilities (PSYDs) to have satisfactory sexuality, to get married, and to adopt a child. The light gray columns with black labels represent the agreement responses. The dark gray columns with white labels represent the disagreement responses. The complete disagreement is represented in black. Labels are presented only for response options chosen by at least 2% of participants.

**Table 1 ijerph-19-01017-t001:** Descriptive statistics.

Sample Characteristics	Variable Levels	Frequencies (%)
Sex	Male	281 (28.9)
	Female	692 (71.1)
Age	18–29	348 (35.8)
	30–39	338 (34.7)
	40 or more	287 (29.5)
Provenience	North	390 (40.1)
	Center	314 (32.3)
	South and Islands	269 (27.6)

Education	Undergraduate	332 (34.1)
	Graduate	391 (40.2)
	Post-graduate	250 (25.7)
Occupation	Student	206 (21.2)
	Employed	620 (63.7)
	Unemployed or retired	147 (15.1)
Relational status	Single	285 (29.3)
	In a committed relationship	260 (26.7)
	Cohabiting with partner	190 (19.5)
	Married	238 (24.5)

Religiosity	Atheist/agnostic	464 (47.7)
	Believer	361 (37.2)
	Churchgoer	148 (15.2)
Sexual orientation	Heterosexual	730 (75.0)
	Not-completely heterosexual, homosexual, bisexual or queer	243 (25.0)

**Table 2 ijerph-19-01017-t002:** Socio-demographic features and percentages of agreement with the right of people with physical and psychical disabilities to have satisfactory sexuality: Frequencies (in percentages) and results of the univariate and multiple logistic regression analyses. Significant differences with respect to the reference are reported in bold.

Agreement with the Right of People with Disabilities to Have Satisfactory Sexuality		People with Physical Disabilities					People with Psychical Disabilities				
		Freq %	Univariate LR		Multiple LR		Freq %	Univariate LR		Multiple LR	
			OR (95% CI)	*p*	OR (95% CI)	*p*		OR (95% CI)	*p*	OR (95% CI)	*p*
Sex	Male	92.5	-	-	-	-	81.9	-	-	-	-
	Female	96.1	**1.99 (1.11–3.58)**	**0.022**	**2.11 (1.11–4.01)**	**0.022**	87.1	**1.5 (1.03–2.19)**	**0.034**	**1.55 (1.03–2.35)**	**0.038**
Age	18–29	94.5	-	-	-	-	86.5	-	-	-	-
	30–39	95.0	1.09 (0.55–2.13)	0.801	0.76 (0.29–1.99)	0.579	84.9	0.88 (0.57–1.35)	0.554	0.84 (0.49–1.47)	0.551
	40 or more	95.8	1.32 (0.63–2.78)	0.458	1.40 (0.48–4.13)	0.538	85.4	0.91 (0.58–1.43)	0.684	1.16 (0.63–2.12)	0.642
Provenience	North	96.2	-	-	-	-	88.2	-	-	-	-
	Center	94.3	0.66 (0.33–1.33)	0.242	0.59 (0.28–1.24)	0.164	86.0	0.82 (0.53–1.28)	0.382	0.74 (0.46–1.19)	0.221
	South and Islands	94.4	0.68 (0.33–1.41)	0.298	0.80 (0.37–2.01)	0.584	81.4	**0.59 (0.38–0.90)**	**0.016**	0.63 (0.39–1.00)	0.051
Instruction	Undergraduate	93.1	-	-	-	-	81.6	-	-	-	-
	Graduate	96.9	**2.35 (1.15–4.80)**	**0.019**	1.91 (0.89–4.08)	0.096	88.5	**1.73 (1.14–2.63)**	**0.010**	**1.73 (1.1–2.72)**	**0.019**
	Post-graduate	94.8	1.36 (0.68–2.74)	0.393	0.89 (0.39–2.01)	0.777	86.4	1.43 (0.91–2.26)	0.124	1.42 (0.85–2.36)	0.179
Occupation	Unemployed-retired	92.5	-	-	-	-	76.9	-	-	-	-
	Employed	96.3	**2.10 (1.00–4.41)**	**0.050**	2.16 (0.97–4.78)	0.058	86.8	**1.97 (1.26–3.09)**	**0.003**	**1.92 (1.18–3.11)**	**0.008**
	Student	93.2	1.11 (0.49–2.52)	0.804	0.95 (0.34–2.67)	0.924	88.3	**2.28 (1.29–4.05)**	**0.005**	**2.47 (1.27–4.81)**	**0.008**
Heterosexual	Yes	94.9	-	-	-	-	84.5	-	-	-	-
	Not completely/no	95.5	1.13 (0.57–2.24)	0.736	1.44 (0.66–3.15)	0.361	88.9	1.47 (0.94–2.29)	0.094	1.35 (0.82–2.22)	0.241
Relational status	Single/casual partners	94.5	-	-	-	-	80.7	-	-	-	-
	In a committed relationship	97.9	**2.33 (1.05–5.16)**	**0.037**	2.64 (0.8–8.75)	0.112	87.3	**1.64 (1.03–2.63)**	**0.037**	1.52 (0.93–2.48)	0.094
	Cohabiting with partner	96.5	**3.89 (1.32–11.47)**	**0.014**	1.86 (0.69–5.04)	0.220	90.0	**2.15 (1.23–3.76)**	**0.007**	**1.92 (1.06–3.48)**	**0.032**
	Married	92.3	1.45 (0.71–2.94)	0.306	0.82 (0.36–1.87)	0.637	86.1	1.49 (0.93–2.38)	0.099	**1.87 (1.11–3.17)**	**0.019**
Religiosity	Atheist/agnostic	95.5	-	-	-	-	89.9	-	-	-	-
	Believer	94.5	0.81 (0.43–1.51)	0.808	0.77 (0.39–1.52)	0.450	82.5	**0.53 (0.36–0.8)**	**0.002**	**0.55 (0.36–0.85)**	**0.007**
	Churchgoer	93.3	0.96 (0.40–2.30)	0.918	0.97 (0.38–2.47)	0.945	79.7	**0.44 (0.27–0.73)**	**0.001**	**0.44 (0.26–0.77)**	**0.004**

LR: logistic regression model; Freq %: percentages; OR: odds ratio; 95 % CI: 95 % confidence interval; -: reference value; Not completely/no: not-completely heterosexual, homosexual, bisexual, or queer.

**Table 3 ijerph-19-01017-t003:** Socio-demographic features and percentages of agreement with the right of people with physical and psychical disabilities to get married: Frequencies (in percentages) and results of the univariate and multiple logistic regression analyses. Significant differences with respect to the reference are reported in bold.

Agreement with the Right of People with Disabilities to Get Married		People with Physical Disabilities					People with Psychical Disabilities				
		Freq %	Univariate LR		Multiple LR		Freq %	Univariate LR		Multiple LR	
			OR (95% CI)	*p*	OR (95% CI)	*p*		OR (95% CI)	*p*	OR (95% CI)	*p*
Sex	Male	97.2	-	-	-	-	71.2	-	-		-
	Female	98.7	2.22 (0.85–5.82)	0.104	2.05 (0.71–5.9)	0.183	81.8	**1.82 (1.32–2.51)**	**0.000**	**1.83 (1.28–2.63)**	**0.001**
Age	18–29	100.0	-	-	-	-	86.2	-	-	-	-
	30–39	97.6	-	-	-	-	76.3	**0.52 (0.35–0.77)**	**0.001**	0.62 (0.37–1.03)	0.066
	40 or more	96.9	**0.36 (0.14–0.95)**	**0.040**	0.9 (0.28–2.85)	0.852	72.5	**0.42 (0.28–0.63)**	**0.000**	0.68 (0.39–1.16)	0.152
Provenience	North	97.9	-	-	-	-	78.7	-	-	-	-
	Center	98.7	1.62 (0.48–5.44)	0.433	1.13 (0.3–4.18)	0.856	81.8	1.22 (0.84–1.77)	0.302	1.03 (0.69–1.55)	0.869
	South and Islands	98.1	1.11 (0.36–3.42)	0.861	0.96 (0.28–3.27)	0.952	75.1	0.82 (0.56–1.18)	0.276	0.76 (0.51–1.14)	0.182
Instruction	Undergraduate	97.3	-	-	-	-	74.4	-	-	-	-
	Graduate	98.7	2.15 (0.71–6.48)	0.174	1.45 (0.42–4.99)	0.557	80.6	**1.43 (1–2.03)**	**0.048**	1.35 (0.91–1.99)	0.134
	Post-graduate	98.8	2.29 (0.61–8.56)	0.217	2.06 (0.49–8.6)	0.321	81.6	**1.53 (1.02–2.29)**	**0.040**	**1.90 (1.21–2.98)**	**0.005**
Occupation	Unemployed-retired	96.6	-	-	-	-	73.5	-	-	-	-
	Employed	98.2	1.95 (0.67–5.7)	0.223	1.9 (0.6–5.98)	0.272	76.6	1.18 (0.78–1.78)	0.423	1.32 (0.84–2.07)	0.224
	Student	99.5	7.22 (0.83–62.45)	0.073	4.49 (0.42–47.69)	0.213	88.8	**2.87 (1.63–5.07)**	**0.000**	**2.31 (1.2–4.43)**	**0.012**
Heterosexual	Yes	97.8	-	-	-	-	75.5	-	-	-	-
	Not completely/no	99.6	5.42 (0.72–41.11)	0.102	2.37 (0.27–20.54)	0.434	88.5	**2.49 (1.63–3.83)**	**0.000**	**1.81 (1.13–2.88)**	**0.013**
Relational status	Single/casual partners	97.5	-	-	-	-	80.0	-	-	-	-
	In a committed relationship	99.6	6.52 (0.8–53.37)	0.080	4.96 (0.58–42.2)	0.143	80.4	1.02 (0.67–1.56)	0.910	0.81 (0.52–1.28)	0.373
	Cohabiting with partner	99.5	4.76 (0.58–39)	0.146	3.39 (0.38–29.87)	0.271	83.7	1.28 (0.79–2.08)	0.312	1.12 (0.66–1.88)	0.680
	Married	96.6	0.72 (0.26–2.03)	0.538	1.18 (0.39–3.63)	0.768	71.4	**0.63 (0.42–0.94)**	**0.023**	0.93 (0.59–1.47)	0.765
Religiosity	Atheist/agnostic	99.6	-	-	-	-	84.9	-	-	-	-
	Believer	97.5	**0.17 (0.04–0.79)**	**0.024**	0.22 (0.04–1.08)	0.062	76.7	**0.59 (0.41–0.83)**	**0.003**	**0.68 (0.46–0.99)**	**0.046**
	Churchgoer	95.9	**0.1 (0.02–0.51)**	**0.006**	**0.14 (0.03–0.76)**	**0.023**	64.2	**0.32 (0.21–0.49)**	**0.000**	**0.38 (0.24–0.61)**	**0.000**

LR: logistic regression model; Freq %: percentages; OR: odds ratio; 95 % CI: 95 % confidence interval; -: reference value; Not completely/no: not-completely heterosexual, homosexual, bisexual, or queer.

**Table 4 ijerph-19-01017-t004:** Socio-demographic features and percentages of agreement with the right of people with physical and psychical disabilities to adopt a child: frequencies (in percentages) and results of the univariate and multiple logistic regression analyses. Significant differences with respect to the reference are reported in bold.

Agreement with the Right of People with Disabilities to Adopt a Child		People with Physical Disabilities					People with Psychical Disabilities				
		Freq %	Univariate LR		Multiple LR		Freq %	Univariate LR		Multiple LR	
			OR (95% CI)	*p*	OR (95% CI)	*p*		OR (95% CI)	*p*	OR (95% CI)	*p*
Sex	Male	81.1	-	-	-	-	19.9	-	-	-	-
	Female	81.9	1.05 (0.74–1.51)	0.771	0.99 (0.67–1.46)	0.949	22.5	1.17 (0.83–1.65)	0.371	1.05 (0.73–1.52)	0.800
Age	18–29	81.6	-	-	-	-	25.9	-	-	-	-
	30–39	84.0	1.19 (0.8–1.76)	0.402	1.29 (0.77–2.14)	0.332	20.7	0.75 (0.52–1.07)	0.111	0.84 (0.53–1.34)	0.465
	40 or more	79.1	0.85 (0.58–1.26)	0.426	1.17 (0.68–2.00)	0.576	18.1	**0.63 (0.43–0.93)**	**0.020**	0.83 (0.49–1.40	0.486
Provenience	North	82.6	-	-	-	-	21.8	-	-	-	-
	Center	84.1	1.12 (0.75–1.66)	0.593	1.04 (0.68–1.58)	0.853	23.9	1.13 (0.79–1.6)	0.511	1.07 (0.74–1.55)	0.712
	South and Islands	77.7	0.74 (0.5–1.08)	0.121	0.74 (0.49–1.11)	0.144	19.3	0.86 (0.58–1.27)	0.444	0.86 (0.58–1.29)	0.479
Instruction	Undergraduate	77.4	-	-	-	-	17.8	-	-	-	-
	Graduate	82.6	1.39 (0.96–2)	0.081	1.37 (0.92–2.04)	0.118	22.8	1.36 (0.94–1.97)	0.098	1.43 (0.97–2.11)	0.074
	Post-graduate	86.0	**1.79 (1.15–2.78)**	**0.009**	**1.84 (1.14–2.98)**	**0.013**	25.6	**1.59 (1.07–2.37)**	**0.023**	**1.97 (1.26–3.08)**	**0.003**
Occupation	Unemployed-retired	77.6	-	-	-	-	21.8	-	-	-	-
	Employed	82.1	1.33 (0.86–2.06)	0.206	1.14 (0.71–1.81)	0.590	19.5	0.87 (0.56–1.35)	0.539	0.82 (0.51–1.30)	0.395
	Student	83.5	1.46 (0.86–2.5)	0.162	1.51 (0.82–2.78)	0.186	28.6	1.44 (0.88–2.37)	0.147	1.51 (0.87–2.65)	0.146
Heterosexual	Yes	79.2	-	-	-	-	20.3	-	-	-	-
	Not completely/no	89.3	**2.19 (1.41–3.42)**	**0.001**	**2.03 (1.26–3.27)**	**0.004**	26.3	**1.41 (1–1.97)**	**0.048**	1.16 (0.8–1.68)	0.430
Relational status	Single/casual partners	79.3	-	-	-	-	20.4	-	-	-	-
	In a committed relationship	83.1	1.28 (0.83–1.98)	0.261	1.28 (0.82–2.01)	0.280	21.9	1.1 (0.73–1.66)	0.653	0.98 (0.64–1.50)	0.917
	Cohabiting with partner	87.9	**1.90 (1.13–3.19)**	**0.016**	1.7 (0.98–2.96)	0.061	26.8	1.44 (0.93–2.21)	0.100	1.42 (0.89–2.26)	0.145
	Married	78.2	0.93 (0.61–1.42)	0.749	1.05 (0.66–1.68)	0.835	19.3	0.94 (0.61–1.44)	0.770	1.29 (0.79–2.10)	0.314
Religiosity	Atheist/agnostic	85.6	-	-	-	-	26.1	-	-	-	-
	Believer	77.6	**0.58 (0.41–0.83)**	**0.003**	0.72 (0.49–1.05)	0.089	20.2	0.72 (0.52–1)	0.050	0.79 (0.55–1.12)	0.181
	Churchgoer	79.7	0.66 (0.41–1.07)	0.092	0.88 (0.53–1.47)	0.623	12.2	**0.39 (0.23–0.67)**	**0.001**	**0.44 (0.25–0.77)**	**0.004**

LR: logistic regression model; Freq %: percentages; OR: odds ratio; 95 % CI: 95 % confidence interval; -: reference value; Not completely/no: not-completely heterosexual, homosexual, bisexual or queer.

## Data Availability

The data presented in this study are available on request from the corresponding author. The Scientific Committee of the Italian Federation of Sexology reserves the right to verify Ethic approval and permissions for research submitted by the applicant, and to approve database sharing at their own discretion.

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
