# Peer review of "The Sexual and Parenting Rights of People with Physical and Psychical Disabilities: Attitudes of Italians and Socio-Demographic Factors Involved in Recognition and Denial"

_ijerph, 2022, doi:10.3390/ijerph19021017_

Round 1

Reviewer 1 Report

Dear Authors,

Thanks for revising the manuscript, which has been greatly improved.

I have no additional concerns. Please fix minor mistkaes (such as references missing a space before the open bracket).

Author Response

Dear Reviewer 1,

many thanks for the time and attention devoted to reviewing our manuscript, and for your positive judgment. Minor mistakes in style and grammar were fixed, according to your suggestions.

Sincerely,

Dr. Simona Gabriella Di Santo and Dr. Margherita Colombo

Reviewer 2 Report

Thank you for your submission and efforts to prepare this manuscript. This article seeks to explore the attitudes of Italians towards the sexual and parenting rights of people with physical and psychical disabilities. Although your manuscript has great merit and is an interesting topic, it needs a little improvement. My comments are as follows:

  1. The manuscript should be edited in terms of English grammar. The manuscript will benefit from general proofreading that addresses issues such as word choice/order, subject-verb agreement, tense, consistency, clarity, and sentence structure etc.
  2. The title of the article can just be simplified as “The attitudes of Italians towards the sexual and parenting rights of people with physical and psychical disabilities
  3. In the abstract, you started straight with the aims of the study. In one or two sentences, can you please begin the abstract with some contextual background to describe the current knowledge and gaps in knowledge regarding the research aims/questions?
  4. In the abstract, I will recommend you summarise the aims but then list all the aims before the Materials and Methods.
  5. In the abstract, please provide a short description of how your participants were recruited.
  6. Your introduction is well-written but please avoid using strong words such as “affected” in line 44 and adopt first-person language – a population with disabilities.
  7. In the Materials and Methods, overall, procedures and data analysis were well described but a bit disorganized. Please it will be nice to make some reorganization on this section: please the ideas are a bit disorganized and I would like to see them organize under the following headings:
  • Study participants
  • Study design
  • Instruments – please describe the instruments for the data collection and how you developed them
  • Data collection – please describe how were the participants recruited and the data collection procedure
  • Data analysis
  1. Please I would like to see some description of the policy implications of these findings.

Author Response

Dear Reviewer 2,

we really appreciate your useful comments and indications, which helped us improve our manuscript. We agree with most of your comments and we have revised our manuscript accordingly. Nevertheless, a point-by-point response to each comment is provided below, including extensive explanations for our decisions not to adhere to some comments. We hope that you will find our responses to your comments satisfactory.

1. The manuscript should be edited in terms of English grammar. The manuscript will benefit from general proofreading that addresses issues such as word choice/order, subject-verb agreement, tense, consistency, clarity, and sentence structure etc.

According to your indications, the paper has been carefully proofread to improve English grammar and readability.

2. The title of the article can just be simplified as “The attitudes of Italians towards the sexual and parenting rights of people with physical and psychical disabilities”.

We agree with you that a simple title may be more readable than a complex one. The title we have chosen, however, with a few more words, is able to completely describe the three aims and the content of our paper. A researcher interested in reviewing all studies that specifically analyzed people with mental and physical disabilities would be able to identify our study as appropriate to his research only by inspecting the title. The same would happen with a researcher wishing to collect specific information on socio-demographic factors associated with the recognition of sexual rights. A shorter and more generic title such as "The attitudes of Italians towards the sexual and parenting rights of people with physical and psychical disabilities" could lead one to mistakenly believe that the paper concerns aspects typical or exclusive of the Italian culture, associated with the recognition of rights. Therefore it is our decision not to change it. Hope you will not be bothered by our decision. Should the magazine need one, we will be happy to use your suggested title as the current title.

3. In the abstract, you started straight with the aims of the study. In one or two sentences, can you please begin the abstract with some contextual background to describe the current knowledge and gaps in knowledge regarding the research aims/questions?

We thank you for your valuable suggestion. Some introductory sentences have been added to the abstract. Unfortunately, we could not provide complete and exhaustive information because the abstract cannot exceed 200 words.

4. In the abstract, I will recommend you summarise the aims but then list all the aims before the Materials and Methods.

We apologize to you, but we did not understand your request: the aims were already listed before the Material and Methods in the abstract. If your advice is to write both general aims and specific aims, we apologize, but due to the limit of 200 words, we cannot accommodate your request. Otherwise, we would have to sacrifice other sentences, in the introduction, in the methods, in the results, or in the conclusions, which instead are fundamental for the completeness of the abstract.

5. In the abstract, please provide a short description of how your participants were recruited.

According to your request, a short description of the sampling method was included in the abstract. 

6. Your introduction is well-written but please avoid using strong words such as “affected” in line 44 and adopt first-person language – a population with disabilities.

According to your indications, the term "is affected" was replaced with "lives with". However, we do not completely agree with your suggestion to consider people with one (or more) disabilities as 'a population with disabilities'. This term may suggest ideas of separation and diversity and could be considered almost disability-phobic. People are people and some of them, among their peculiar characteristics that make them unique and different from others, also have the characteristic of living with one or more impairments within environments that create more or less passable barriers to the full realization of their rights. Consistent with the UNCRPD vision that there is no 'population with disabilities', but that disabilities are determined by the specific geographical, cultural, socio-economic, and other-related-variable context, in which each person with disabilities lives, we prefer to continue using the term in the plural (people with). We hope you do not mind. 

7. In the Materials and Methods, overall, procedures and data analysis were well described but a bit disorganized. Please it will be nice to make some reorganization on this section: please the ideas are a bit disorganized and I would like to see them organize under the following headings:

  • Study participants
  • Study design
  • Instruments – please describe the instruments for the data collection and how you developed them
  • Data collection – please describe how were the participants recruited and the data collection procedure
  • Data analysis

The subsections of materials and methods have been reorganized according to your requests, modifying where possible the subheadings as per your indication, and, in other cases, mediating between your indications and the original subheadings, for greater clarity.

  1. Please I would like to see some description of the policy implications of these findings.

According to your requests, a brief description of the policy implications of these findings has been added in the conclusion section.

We hope that the revised version has satisfactorily addressed all issues.

Sincerely,

Dr. Simona Gabriella Di Santo and Dr. Margherita Colombo